*Scientific Report*

# Miro1-dependent mitochondrial positioning drives the rescaling of presynaptic Ca$^{2+}$ signals during homeostatic plasticity

Victoria Vaccaro[†], Michael J Devine[†], Nathalie F Higgs[†] & Josef T Kittler[*] (iD)

## Abstract

**Mitochondrial trafficking is influenced by neuronal activity, but it remains unclear how mitochondrial positioning influences neuronal transmission and plasticity. Here, we use live cell imaging with the genetically encoded presynaptically targeted Ca$^{2+}$ indicator, SyGCaMP5, to address whether presynaptic Ca$^{2+}$ responses are altered by mitochondria in synaptic terminals. We find that presynaptic Ca$^{2+}$ signals, as well as neurotransmitter release, are significantly decreased in terminals containing mitochondria. Moreover, the localisation of mitochondria at presynaptic sites can be altered during long-term activity changes, dependent on the Ca$^{2+}$-sensing function of the mitochondrial trafficking protein, Miro1. In addition, we find that Miro1-mediated activity-dependent synaptic repositioning of mitochondria allows neurons to homeostatically alter the strength of presynaptic Ca$^{2+}$ signals in response to prolonged changes in neuronal activity. Our results support a model in which mitochondria are recruited to presynaptic terminals during periods of raised neuronal activity and are involved in rescaling synaptic signals during homeostatic plasticity.**

**Keywords** homeostatic; Miro1; mitochondria; plasticity; synapse
**Subject Categories** Membrane & Intracellular Transport; Neuroscience

## Introduction

Mitochondria play an important role in maintaining neuronal function due to their ability to produce the energy substrate ATP and to buffer local Ca$^{2+}$ rises [1–3]. Presynaptic Ca$^{2+}$ signals trigger vesicular release and their amplitude is known to influence synaptic transmission [4,5]. Previous pharmacological studies suggest that mitochondria can efficiently buffer Ca$^{2+}$ in presynaptic terminals [6,7]. However, only a subset of presynaptic terminals within the same axon may contain mitochondria [8], and the impact of mitochondrial occupancy on presynaptic Ca$^{2+}$ signalling and vesicular release in individual terminals within the same axon remains poorly understood.

Homeostatic plasticity plays a central role in stabilising network activity by rescaling synaptic weights and neuronal excitability in accordance with the activity level of neurons [9]. The homeostatic rescaling of the efficiency of synapses in order to avoid extreme levels of activity is dependent on changes in both presynaptic and postsynaptic function [9–11]. Interestingly, recent developments in imaging Ca$^{2+}$ signals using genetically encoded presynaptically targeted Ca$^{2+}$ indicators have shown that presynaptic Ca$^{2+}$ responses undergo homeostatic plasticity [12]. It is unclear however how the reported rescaling of presynaptic Ca$^{2+}$ signals is mediated and whether mitochondrial Ca$^{2+}$ buffering can play a role in homeostatic changes of neuronal transmission efficiency.

Mitochondrial positioning can be regulated by neuronal activity, dependent on the mitochondrial trafficking protein Miro1 [13–15]. Miro1 is located in the outer mitochondrial membrane and contains two Ca$^{2+}$-sensing EF-hand domains by which it responds to local Ca$^{2+}$ signalling to interrupt mitochondrial trafficking, thus depositing mitochondria at subcellular locations of high activity [13,14,16]. However, whether mitochondrial positioning at synapses is altered during long-term changes in neuronal activity and whether a role exists for Miro1-mediated mitochondrial trafficking in the tuning of synaptic mitochondrial occupancy remains unclear.

Here, by imaging presynaptic Ca$^{2+}$ [17] and mitochondrial positioning in multiple terminals of the same axon, we show that mitochondrial occupancy determines presynaptic Ca$^{2+}$ responses and can also affect vesicular release. Moreover, we demonstrate that mitochondrial localisation at presynaptic terminals is tuned by long-term changes in network activity dependent on Miro1-mediated mitochondrial trafficking. Further, we show that baseline Ca$^{2+}$ responses and homeostatic changes in the presynaptic response are altered in the absence of Miro1-mediated Ca$^{2+}$-dependent positioning of mitochondria. This provides evidence for a novel mechanism by which mitochondria can alter presynaptic transmission and play a role in the tuning of synaptic signals during homeostatic plasticity.

Department of Neuroscience, Physiology and Pharmacology, University College London, London, UK
*Corresponding author. Tel: +44 020 7679 3218; E-mail: j.kittler@ucl.ac.uk
†These authors contributed equally to this work

# Results and Discussion

In order to investigate a potential difference in the $Ca^{2+}$ signals evoked in presynaptic terminals containing mitochondria compared to terminals without mitochondria, we co-transfected hippocampal cultures with the mitochondrial marker MtDsRed and the presynaptically targeted version of the genetically encoded $Ca^{2+}$ indicator GCaMP5 (based on SyGCaMP2 and GCaMP5 [17,18]). Labelling of SyGCaMP5-transfected neurons with the presynaptic markers SV2 and Piccolo confirmed that the indicator is presynaptically targeted (Fig EV1) and, using this approach, terminals with mitochondria could be easily distinguished from those without by merging both acquisition channels (Fig 1A).

While stimulating neurons electrically, with extracellular field electrodes at 10 Hz for 10 s (thus generating 100 action potentials (APs); see Materials and Methods), we compared the average presynaptic $Ca^{2+}$ signal and found that, in terminals without mitochondria, the average $Ca^{2+}$ signal during the time of stimulation ($t = 20$–$30$ s) was significantly greater ($\Delta F/F_0 = 3.5 \pm 0.4$) than in terminals containing mitochondria ($\Delta F/F_0 = 1.9 \pm 0.3$, $n = 11$ neurons, 91 terminals, $P < 0.001$; Fig 1B and C). Importantly, the mean stimulation $\Delta F/F_0$ is not determined by the unnormalised baseline fluorescence within each terminal ($r = 0.05$, $P > 0.2$ for $n = 90$ terminals; Fig EV2A). Further, there is no significant difference between the baseline fluorescence signals in those terminals occupied with a mitochondrion compared to those without ($P = 0.1$; Fig EV2A). In contrast, upon a very brief stimulation of 1 ms, which should only lead to initiation of a single AP (Fig 1D and E), we did not observe a significant difference in the presynaptic $Ca^{2+}$ responses when those terminals occupied by a mitochondrion were compared to terminals in the same axon not occupied by a mitochondrion (Fig 1D and E).

Therefore, we sought to determine the threshold number of stimuli that was required to elicit a significant difference in presynaptic $Ca^{2+}$ response in the presence of a mitochondrion. We found that a train of 10 stimuli (10 APs) was insufficient to generate a difference in $Ca^{2+}$ response ($\Delta F/F_0 = 0.4 \pm 0.3$ with and $0.4 \pm 0.1$ without mitochondria, $n = 3$ neurons, 21 terminals, $P = 0.90$; Fig 1F). This was verified in a separate data set obtained at maximum acquisition frame rate of ~18 frames per second ($\Delta F/F_0 = 1.0 \pm 0.2$ with and

$1.2 \pm 0.3$ without mitochondria, $n = 13$ neurons, 44 terminals, $P = 0.68$; Fig EV2B), to exclude the possibility that transient differences were missed when data were acquired at 1 frame per second. In contrast, trains of stimuli $\geq 20$ were sufficient to generate a difference in presynaptic $Ca^{2+}$ response (20 stimuli: $\Delta F/F_0 = 0.8 \pm 0.4$ with and $1.8 \pm 0.4$ without mitochondria, $n = 10$ neurons, 55 terminals, $*P < 0.05$; 40 stimuli: $\Delta F/F_0 = 0.9 \pm 0.3$ with and $2.0 \pm 0.4$ without mitochondria, $n = 5$ neurons, 25 terminals, $*P < 0.05$; 80 stimuli: $\Delta F/F_0 = 1.3 \pm 0.2$ with and $2.2 \pm 0.5$ without mitochondria, $n = 7$ neurons, 21 terminals, $*P < 0.05$; Fig 1F). Next, we varied the frequency at which these stimuli were delivered, to see whether the rate of rise in presynaptic $Ca^{2+}$ would have an impact on the ability of mitochondria to buffer this signal. The differences observed at 20 stimuli were maintained across a range of frequencies (5 Hz: $\Delta F/F_0 = 0.9 \pm 0.2$ with and $1.3 \pm 0.4$ without mitochondria, $n = 11$ neurons, 58 terminals, $*P < 0.05$; 100 Hz: $\Delta F/F_0 = 1.2 \pm 0.3$ with and $1.7 \pm 0.4$ without mitochondria, $n = 12$ neurons, 87 terminals, $*P < 0.05$; Fig EV2B). In contrast, no differences were observed with 10 stimuli delivered at 100 Hz ($\Delta F/F_0 = 0.8 \pm 0.2$ with and $0.8 \pm 0.2$ without mitochondria, $n = 4$ neurons, 18 terminals, $P = 0.60$; Fig EV2B), despite the more rapid delivery of APs and thus more rapid rise in presynaptic $Ca^{2+}$.

These findings suggest that even though single APs can generate small mitochondrial $Ca^{2+}$ transients [19], mitochondria are more relevant as $Ca^{2+}$ buffers after prolonged stimulation. This agrees with previous studies suggesting that mitochondria play a role after $Ca^{2+}$ has accumulated in the presynaptic terminal [20,21] and is also supported by earlier less direct studies which demonstrate that either pharmacological inhibition of mitochondrial $Ca^{2+}$ buffering or genetic intervention to direct mitochondria out of terminals increases presynaptic $Ca^{2+}$ responses [6,22–25]. The threshold of 20 stimuli could correspond to depletion of the entire readily releasable pool of neurotransmitter vesicles, because this is thought to be fully released within 2 s of 20 Hz stimulation [26].

We then asked whether the observed difference in $Ca^{2+}$ response is due to mitochondrial $Ca^{2+}$ uptake mediated by the mitochondrial $Ca^{2+}$ uniporter (MCU) and thus used 30 min Ru360 treatment to block its activity. In the control cultures, presynaptic terminals containing a mitochondrion had a decreased mean $Ca^{2+}$ signal

**Figure 1. Mitochondrial occupancy decreases presynaptic $Ca^{2+}$ signals.**

A    Live images of neurons co-transfected with SyGCaMP5 and MtDsRed before and during 10 Hz field stimulation. The full arrow indicates a mitochondrially occupied terminal and the empty arrow a terminal without a mitochondrion. Scale bar, 10 μm.

B    Average $\Delta F/F_0$ SyGCaMP5 traces from $n = 11$ neurons (98 terminals) plotting the average of terminals without a mitochondrion (black trace) and with a mitochondrion (red trace). Stimulation occurred for 10 s ($t = 20$–$30$) at 10 Hz.

C    Average $Ca^{2+}$ response following stimulation in terminals with or without a mitochondrion (average of $\Delta F$ measurement taken for a $\Delta t = 20$–$30$ s of B), $\Delta F/F_0 = 1.9 \pm 0.3$ with and $3.5 \pm 0.4$ without mitochondria, paired $t$-test, $***P < 0.0001$.

D, E    There is no significant difference in the presynaptic $Ca^{2+}$ response in terminals with and without a mitochondrion in response to single action potentials. (D) Example trace of hippocampal neurons transfected with MtDsRed and SyGCaMP5. 1 ms stimulation pulses were applied every 5 s. The red trace represents an average of the terminals occupied with a mitochondrion in this neuron, whereas the black trace represents the averaged response of terminals without a mitochondrion. (E) Maximal fluorescence intensity in terminals without a mitochondrion (left) or with a mitochondrion (right) within the same axon (dots connected by a line) ($n = 8$ neurons, 44 terminals, paired $t$-test, $P = 0.55$).

F    Average $\Delta F/F_0$ SyGCaMP5 traces for 80 stimuli ($n = 7$ neurons, 21 terminals, $\Delta t = 20$–$28$ s, $\Delta F/F_0 = 1.3 \pm 0.2$ with and $2.2 \pm 0.5$ without mitochondria, paired $t$-test, $*P < 0.05$), 40 stimuli ($n = 5$ neurons, 25 terminals, $\Delta t = 20$–$24$ s, $\Delta F/F_0 = 0.9 \pm 0.3$ with and $2.0 \pm 0.4$ without mitochondria, paired $t$-test, $*P < 0.05$), 20 stimuli ($n = 10$ neurons, 55 terminals, $\Delta t = 20$–$22$ s, $\Delta F/F_0 = 0.8 \pm 0.4$ with and $1.8 \pm 0.4$ without mitochondria, paired $t$-test, $*P < 0.05$) and 10 stimuli ($n = 4$ neurons, 21 terminals, $\Delta t = 20$–$21$ s, $\Delta F/F_0 = 0.4 \pm 0.3$ with and $0.4 \pm 0.1$ without mitochondria, paired $t$-test, $P = 0.90$), all delivered at 10 Hz.

Data information: Experiments were performed in E18 or P0 rat hippocampal neuronal cultures at DIV 10–12. Error bars represent SEM.

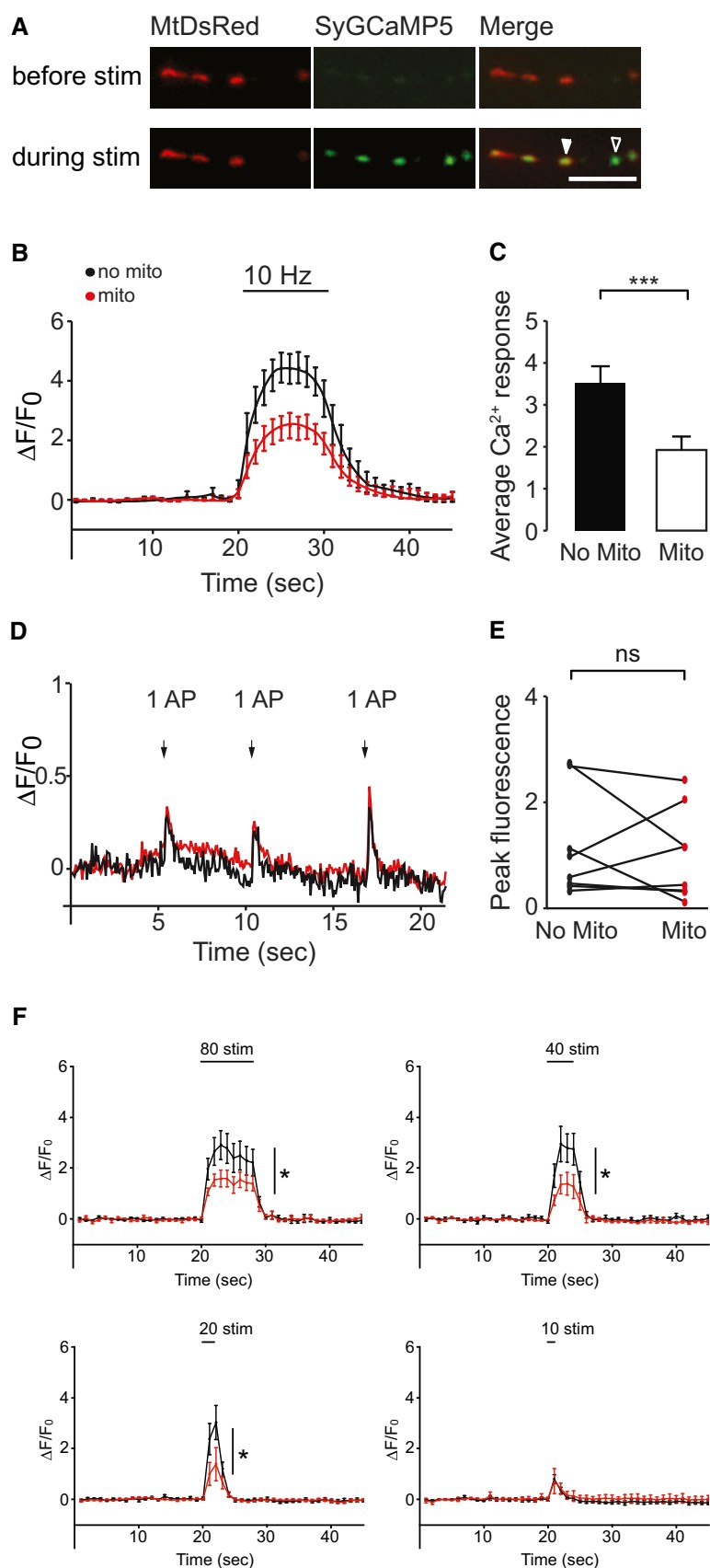

**Figure 1.**

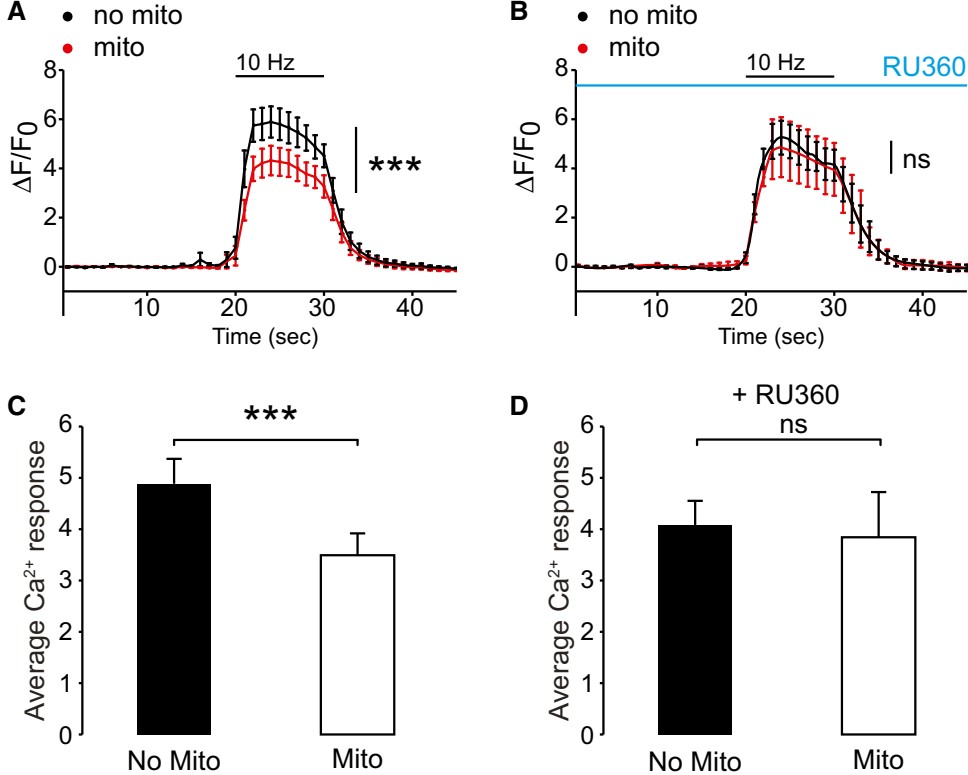

**Figure 2. Presynaptic mitochondrial $Ca^{2+}$ buffering depends upon MCU activity.**

A, B Average trace of hippocampal neurons transfected with MtDsRed and SyGCaMP5 imaged after incubation in ACSF (A) and treatment with 10 μM Ru360 (B) for 30 min. Neurons were stimulated using field stimulation for 10 s at 10 Hz. Red trace = average of the terminals occupied with a mitochondrion; black trace = averaged response of terminals without a mitochondrion ($n$ = 15 neurons, 96 terminals in A; $n$ = 13 neurons, 79 terminals in B; ***$P$ < 0.001, paired $t$-test).

C Summary bar graph of the control neurons (corresponding to the traces shown in panel A). Average $Ca^{2+}$ response ($\Delta t$ = 20–30 s) following stimulation for terminals occupied by a mitochondrion ($\Delta F/F_0$ = 3.5 ± 0.4) and unoccupied terminals ($\Delta F/F_0$ = 4.9 ± 0.5, $n$ = 15 neurons, 96 terminals, ***$P$ < 0.001, paired $t$-test).

D Summary bar graph of the Ru360-treated neurons (corresponding to the traces shown in panel B). Average $Ca^{2+}$ response ($\Delta t$ = 20–30 s) following stimulation for terminals occupied by a mitochondrion ($\Delta F/F_0$ = 4.1 ± 0.5) and unoccupied terminals ($\Delta F/F_0$ = 3.8 ± 0.9, $n$ = 13 neurons, 79 terminals, $P$ = 0.53, paired $t$-test).

Data information: Experiments were performed in P0 rat hippocampal neuronal cultures at DIV 10–12. Error bars represent SEM.

($\Delta F/F_0$ = 3.5 ± 0.4) compared to terminals without a mitochondrion ($\Delta F/F_0$ = 4.9 ± 0.5; Fig 2A and C; $n$ = 15 neurons, 96 terminals, ***$P$ < 0.001). When MCU activity was blocked, the presence of mitochondria no longer affected the mean stimulation $Ca^{2+}$ signal $\Delta F/F_0$ (Fig 2B and D; 4.1 ± 0.5 for mitochondrially occupied terminals and 3.8 ± 0.9 for terminals without a mitochondrion, $n$ = 13 neurons, 79 terminals, $P$ = 0.53). Thus, mitochondria play a key role in the regulation of the size of presynaptic $Ca^{2+}$ signals during trains of action potentials, contingent on their ability to buffer $Ca^{2+}$ in the terminal via MCU.

Having established that mitochondrial occupancy and $Ca^{2+}$ buffering can significantly affect presynaptic $Ca^{2+}$ signals at individual synapses, we determined whether the presence of mitochondria impacts on transmitter release. To image vesicular release, we used the supereliptic pHluorin-based probe, VGlut1pHluorin, while imaging mitochondrial positioning with a mitochondrially targeted variant of LSS-mKate2 (Fig 3A) [27,28]. When stimulating the neurons for 20 s at 10 Hz, we observed a significant increase in the VGlut1pHluorin average signal in terminals without a mitochondrion compared to those terminals occupied by a mitochondrion within the same axon ($\Delta F/F_0$ = 3.2 ± 0.7, terminals without

mitochondria; $\Delta F/F_0$ = 2.3 ± 0.9, terminals with mitochondria; $n$ = 9 neurons, 66 terminals, *$P$ < 0.05; Fig 3B and C). These findings suggest that the presence of mitochondria decreases local $Ca^{2+}$ signals via MCU, leading to less vesicular fusion. While presynaptic $Ca^{2+}$ signalling and vesicular release have previously been shown to be unaffected by MCU knockdown [29], in this study no separation was made between terminals containing or not a mitochondrion, which may have led to an underestimation of the mitochondrial impact on presynaptic $Ca^{2+}$ due to the simultaneous sampling of all synapses.

Next, we sought to establish whether mitochondrial occupancy could determine activity-dependent changes in presynaptic $Ca^{2+}$ signalling, via modulating the size of the presynaptic $Ca^{2+}$ response. Mitochondria have been observed to stop in axons and at postsynaptic terminals in response to local activity [13,14], but whether prolonged changes in neuronal activity lead to a redistribution of mitochondria to and from synapses remains unclear. In order to investigate whether long-term changes in neuronal activity (48 h) alter the recruitment of mitochondria to presynaptic terminals, we co-transfected neurons with synaptophysin–GFP (SYN-GFP) to label presynaptic terminals and MtDsRed to label mitochondria (Fig 4A)

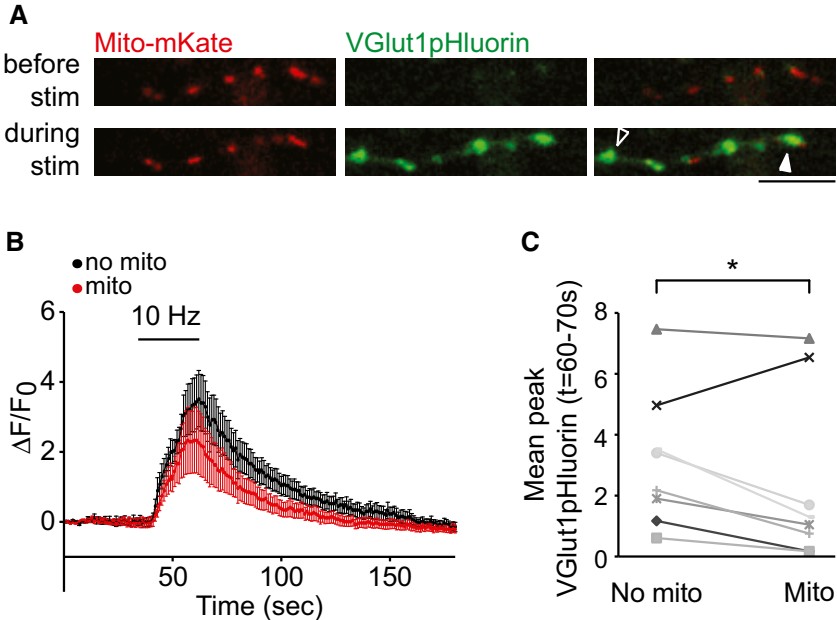

**Figure 3. Vesicular release is reduced in the presence of mitochondria.**

A   Example images of hippocampal neurons co-transfected with Mito-mKate2 and VGlut1pHluorin before and during field stimulation. The white arrow indicates a terminal occupied with a mitochondrion, while the empty arrow indicates a terminal without a mitochondrion. Scale bar, 5 μm.

B   Average traces of presynaptic terminals of hippocampal neurons transfected with VGlut1pHluorin and Mito-mKate2 and stimulated using field stimulation at 10 Hz for 20 s (40–60 s). Terminals with a mitochondrion are represented by the red trace, and terminals without are represented by the black trace.

C   Summary graph corresponds to the traces shown in panel (B). Mean stimulation fluorescence (time points 60–70 s) comparing terminals of the same axon which contain mitochondria (right, $\Delta F/F_0$ = 2.3 ± 0.9) compared to terminals without mitochondria (left, $\Delta F/F_0$ = 3.2 ± 0.7; $n$ = 9 neurons, 66 terminals, *$P$ < 0.05, paired $t$-test).

Data information: Experiments were performed in P0 rat hippocampal neuronal cultures at DIV 10–12. Error bars represent SEM.

and quantified the co-localisation of these two reporters. The resulting fraction of terminals with mitochondria (34.3% ± 3.6) is similar to previous findings [8]. To decrease network activity, we treated neurons with the sodium channel blocker tetrodotoxin (TTX) (1 μM, 48 h), whereas an increase in activity was achieved by applying the GABA$_A$ receptor antagonist picrotoxin (PTX) (100 μM, 48 h) [30]. Silencing neurons with TTX led to a significant decrease in the fraction of SYN-GFP clusters containing a mitochondrion (16.6% ± 3.1, $P$ < 0.01), whereas conversely increasing neuronal activity, driven by PTX treatment, increased the fraction of SYN-GFP synapses containing mitochondria compared to control (DMSO vehicle control 28.7% ± 3.9, PTX 50.0% ± 7.2, **$P$ < 0.01; Fig 4A and D) indicating that long-term activity serves to alter mitochondrial occupancy at the synapse. Importantly, the density of mitochondria (mitochondria/μm: control 0.103 ± 0.01, TTX 0.108 ± 0.01, DMSO 0.126 ± 0.02, PTX 0.127 ± 0.02; $P$ > 0.05) and SYN-GFP clusters (synapse/μm: control 0.218 ± 0.01, TTX 0.205 ± 0.01, DMSO 0.231 ± 0.01, PTX 0.218 ± 0.02; $P$ > 0.05) throughout the axon was unaltered following both TTX and PTX treatments (Fig EV3A and B). Therefore, variations in mitochondrial occupancy of presynaptic terminals are due to changes in the location of mitochondria, rather than an overall change in the number of mitochondria or synapses in the axon, and the redistribution of mitochondria is from a local and previously available pool. A recent paper illustrated that the frequency of short mitochondrial pauses at synapses is increased when neuronal cultures

are stimulated using field stimulation and that a block of neuronal activity using TTX leads to an increase in mitochondrial velocity in the axon [31], which may explain how we come to observe our changes in occupancy after long-term treatment with PTX and TTX, respectively.

To confirm that the mitochondrial impact on vesicular release persists after PTX-induced mitochondrial redistribution, we imaged vesicular release with VGlut1pHluorin with respect to mitochondrial position (with LSS-mKate) following 48 h of PTX treatment. When stimulating neurons for 20 s at 10 Hz, we again detected a significant increase in VGlut1pHluorin signal in terminals without a mitochondrion compared to terminals occupied by a mitochondrion within the same axon ($\Delta F/F_0$ = 3.1 ± 0.3, terminals without mitochondria; $\Delta F/F_0$ = 2.4 ± 0.4, terminals with mitochondria; $n$ = 12 neurons, 58 terminals, *$P$ < 0.05; Fig EV3C and D). Thus mitochondria are capable of downregulating vesicular release following their activity-dependent synaptic recruitment, which could contribute to the homeostatic rescaling of synaptic signals in response to a prolonged elevation of network activity.

To further explore the mechanism of activity-dependent change in mitochondrial occupancy, we asked whether Miro1-mediated trafficking might play a role in long-term alterations in mitochondrial positioning. To evaluate whether the relocation of mitochondria upon prolonged activity changes is dependent on Miro1-mediated stopping of mitochondria at presynaptic sites, we expressed wild-type Miro1 (myc-Miro1) and a Ca$^{2+}$-insensitive mutant, ΔEF-Miro1 [13], to

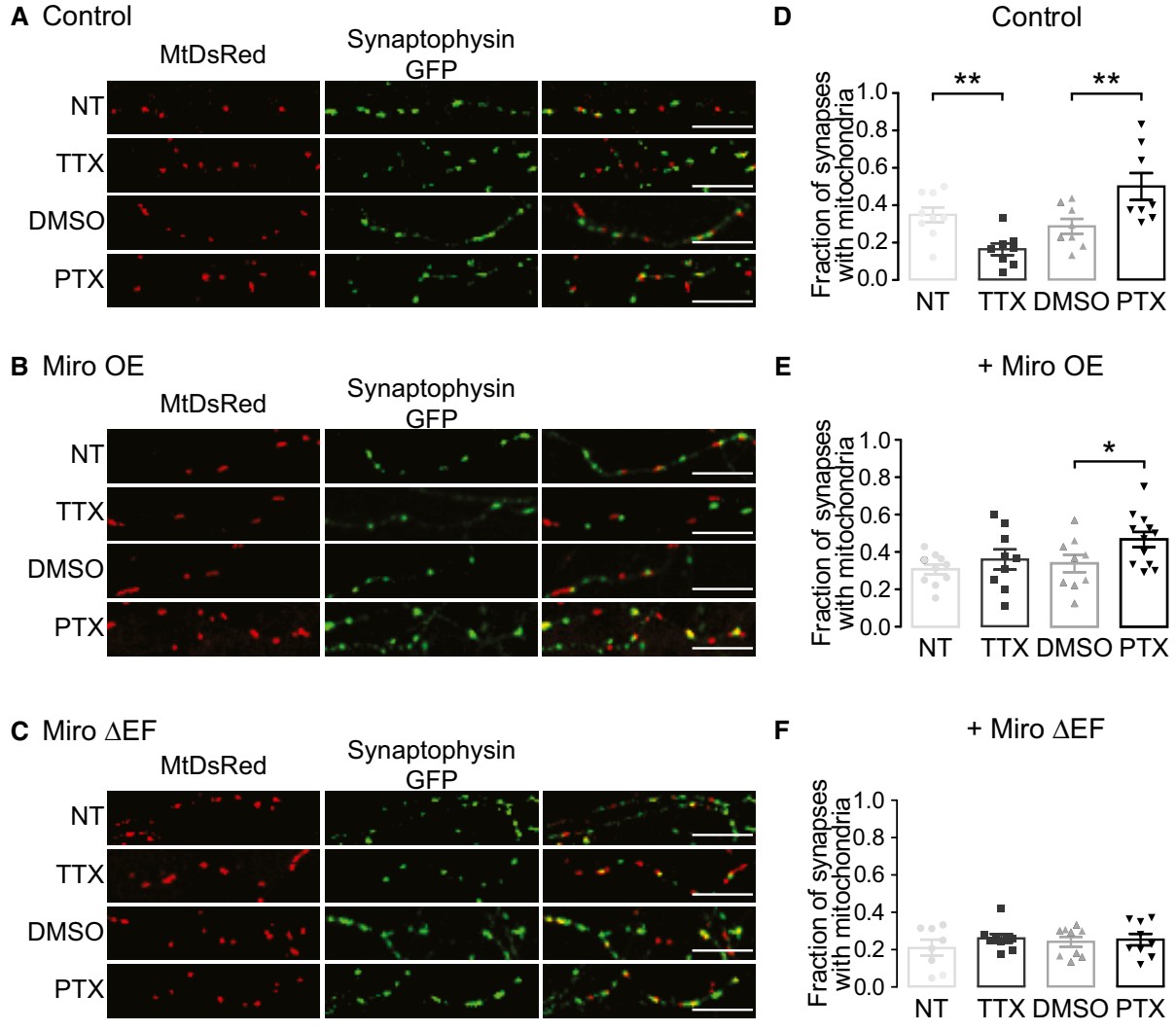

**Figure 4.  Neuronal activity influences mitochondrial occupancy of presynaptic terminals in a Miro1-dependent manner.**

A–C   Confocal images of neuronal processes of fixed neurons transfected with (A) MtDsRed and SYN-GFP and (B) myc-Miro1 (Miro OE) or (C) myc-Miro1ΔEF (Miro ΔEF). Neurons are either non-treated (NT), TTX-treated (1 μM, 48 h), PTX-treated (100 μM, 48 h) or DMSO-treated (1:2,000, as PTX). Scale bars, 10 μm.

D–F   Fraction of SYN-GFP clusters co-localising with mitochondria in cultures in (D) control conditions (DMSO 28.7% ± 3.9, TTX 16.6% ± 3.1, PTX 50.0% ± 7.2, ANOVA **$P < 0.001$, $n = 8$–9 neurons), (E) with the expression of myc-Miro1 (DMSO 33.9% ± 4.7, TTX 35.9% ± 5.4, PTX 46.7% ± 4.0, ANOVA *$P < 0.05$, $n = 8$–9) or (F) with the expression of myc-Miro1ΔEF (ANOVA $P > 0.05$, $n = 9$–12).

Data information: Experiments were performed in E16 mouse hippocampal neuronal cultures at DIV 10–12. Error bars represent SEM.

uncouple constitutive from activity-dependent mitochondrial trafficking (Fig EV4A–C). Overexpression of both Miro1 and ΔEF-Miro1 had no effect on the formation or localisation of SYN-GFP clusters (Fig EV4D and E). However, upon expression of wild-type Miro1, the fraction of SYN-GFP puncta overlapping MtDsRed was no longer decreased after TTX treatment, but PTX treatment still led to an increase in co-localisation (Fig 4B and E). Expression of ΔEF-Miro1 on the other hand suppressed any activity-dependent changes in mitochondrial occupancy upon PTX and TTX treatment (Fig 4C and F). Inactivity induced by glutamate receptor blockade (APV: 100 μM, NBQX: 10 μM, 24 h), known to initiate presynaptic homeostatic plasticity due to a compensatory increase in presynaptic release probability [32], also resulted in a change in mitochondrial presynaptic occupancy (control 0.302 ± 0.02, APV + NBQX

0.214 ± 0.02, *t*-test *$P < 0.05$; Fig EV4F). These activity-dependent changes were again inhibited following the expression of ΔEF-Miro1 (control 0.256 ± 0.03, APV + NBQX 0.259 ± 0.05; Fig EV4H).

Several studies have shown that Miro1 and its two EF-hand domains contribute to $Ca^{2+}$-dependent mitochondrial stopping and recruitment to areas of high $Ca^{2+}$ load [13,14]. We hypothesise that increased expression of Miro1 makes mitochondria more likely to arrest close to presynaptic terminals in response to local $Ca^{2+}$ as Miro1 overexpression increases mitochondrial trafficking [13,14,33] and mitochondria pass terminals more frequently. Thus, even rare $Ca^{2+}$ responses during TTX treatment might be sufficient to allow mitochondrial arrest when Miro1 is overexpressed, and therefore, a higher percentage of terminals are always occupied by a mitochondrion. PTX on the other hand has the same effect whether Miro1 is

overexpressed or not, because during periods of high activity mitochondria are eventually repositioned to presynaptic terminals in response to the frequent $Ca^{2+}$ signals. We therefore propose that mitochondria are arrested at terminals in a Miro1-mediated activity-dependent manner, followed by tethering of the mitochondria by a protein such as syntaphilin [25].

Long-term changes in neuronal activity in hippocampal cultures have been shown to generate homeostatic changes in presynaptic $Ca^{2+}$ responses [12]. As mitochondria can reduce presynaptic $Ca^{2+}$ signals and transmitter release (Fig 1) and long-term activity changes can drive mitochondrial movement into and out of terminals, we hypothesised that Miro1-mediated mitochondrial repositioning may contribute to homeostatic plasticity of the presynaptic $Ca^{2+}$ response. As previously described, neurons were transfected with MtDsRed and SyGCaMP5 and the presynaptic $Ca^{2+}$ response within a region of the axon was measured. TTX treatment (1 µM, 48 h) resulted in the expected increase in presynaptic $Ca^{2+}$ signals, likely due to the fact that less terminals contain a mitochondrion ($1.3 \pm 0.6$ for non-treated cultures, $2.5 \pm 0.5$ for TTX-treated cultures; $n = 11$ and $n = 10$ neurons, 69 and 62 terminals, respectively, *$P < 0.05$; Fig 5B). To address the impact of blocking activity-dependent mitochondrial recruitment to presynaptic terminals, we expressed the $Ca^{2+}$-insensitive mutant of Miro1 (ΔEF-Miro1) and determined whether this interferes with the presynaptic rescaling of $Ca^{2+}$ signals. To facilitate live cell imaging in ΔEF-Miro1-transfected neurons, we co-transfected SyGCaMP5 with ΔEF-Miro1-IRES-MtDsRed, which allows bicistronic expression of both

transgenes thus permitting us to readily identify ΔEF-Miro1 expression in MtDsRed-positive neurons co-expressing SyGCaMP5 (Fig EV5). We found that the homeostatic increase in $Ca^{2+}$ response after TTX treatment was occluded when ΔEF-Miro1 is expressed ($\Delta F/F_0 = 2.3 \pm 0.6$ for non-treated cultures and $\Delta F/F_0 = 1.8 \pm 0.3$ for TTX-treated cultures; $n = 7$ cells, 33 and 32 terminals, respectively, $P = 0.05$; Fig 5B and C) suggesting a role for activity-dependent tuning of mitochondrial presynaptic terminal occupancy in the homeostatic scaling of presynaptic $Ca^{2+}$ signals. This suggests that some of the homeostatic rescaling we observe under normal conditions is dependent on Miro1 function and in particular on Miro1's $Ca^{2+}$-sensing ability mediated via its two EF-hand domains.

Our findings suggest that mitochondrial occupancy of a presynaptic terminal can tune local $Ca^{2+}$ signals via MCU, to regulate vesicular fusion. The probability of release ($P_{rel}$) is known to vary even in terminals from the same axon [34,35]. This can arise from differences in the size of the readily releasable pool of these terminals [34,36] but may also arise from variability in $Ca^{2+}$ dynamics and fusion probability of vesicles [37]. Mitochondrial ATP provision was also recently proposed to contribute to the variability of presynaptic strength [33], particularly during long stimulation trains. However, other studies using presynaptically targeted ATP probes showed that even long stimulation periods of 60 s do not necessarily lead to a depletion of presynaptic ATP, because activity-driven ATP generation (through glycolysis and oxidative phosphorylation) [38] and ATP diffusion [39] can serve to maintain presynaptic ATP levels. Thus, the importance of local mitochondrial ATP provision

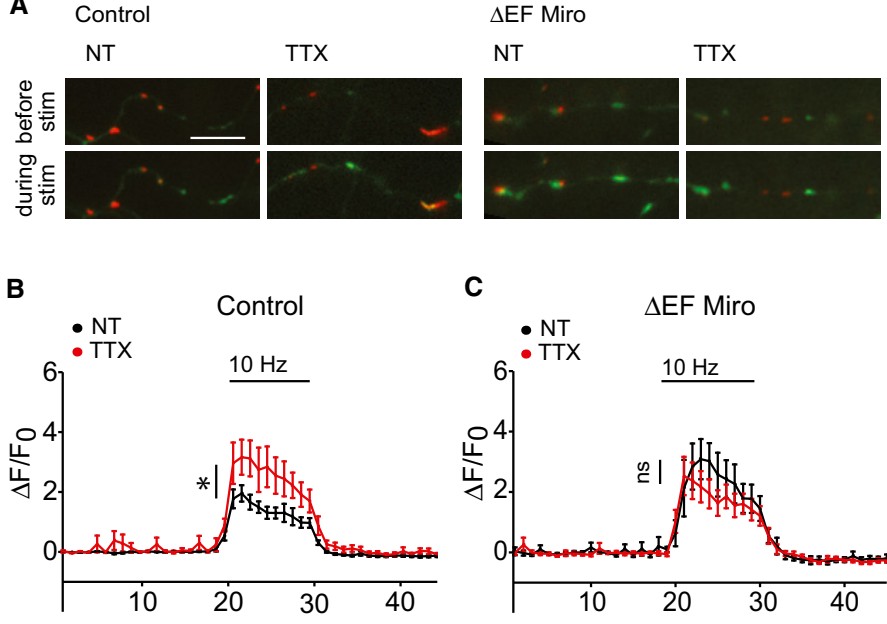

**Figure 5. Miro1 is involved in homeostatic changes in presynaptic $Ca^{2+}$ signals.**

A   Live images of neurons co-transfected with SyGCaMP5 and MtDsRed or myc-ΔEF-Miro1-IRES-MtDsRed (ΔEF Miro) before and during 10-Hz field stimulation with and without TTX treatments. Scale bar, 10 µm.

B   Average $\Delta F/F_0$ SyGCaMP5 traces from terminals treated with TTX (red trace, $\Delta F/F_0 = 2.5 \pm 0.5$) in $n = 10$ neurons (62 terminals) and non-treated terminals (black trace, $\Delta F/F_0 = 1.3 \pm 0.6$) in $n = 11$ neurons (69 terminals) co-transfected with MtDsRed; *$P < 0.05$, t-test.

C   Average $\Delta F/F_0$ SyGCaMP5 traces from terminals treated with TTX (red trace, $\Delta F/F_0 = 1.8 \pm 0.3$) in $n = 7$ neurons (33 terminals) and non-treated terminals (black trace, $\Delta F/F_0 = 2.3 \pm 0.6$) in $n = 7$ neurons (32 terminals) co-transfected with myc-ΔEF-Miro1-IRES-MtDsRed; $P = 0.5$, t-test.

Data information: Experiments were performed in E16 mouse hippocampal neuronal cultures at DIV 10–12. Error bars represent SEM.

at terminals occupied by a mitochondrion in sustaining vesicular release may vary dependent on signalling demands (e.g. duration and frequency of AP firing) and synapse type [19,33,38,39]. Here, we show that mitochondrial occupancy of a terminal can directly impact the size of the $Ca^{2+}$ response upon a train of APs. This is in agreement with a recent report that presynaptic mitochondria in cortical axons attenuate neurotransmitter release by enhancing $Ca^{2+}$ clearance in an LKB1-dependent manner [40]. Thus, $Ca^{2+}$ buffering by mitochondria may be the more important mediator of local mitochondrial influence at presynaptic terminals during synaptic transmission.

Homeostatic changes are important for regulating overall levels of activity in neuronal networks by avoiding extreme states of excitation or inhibition in the brain [9]. The synaptic rescaling during homeostatic plasticity is partly dependent on alteration of AMPA receptor levels at the postsynaptic density [9] but has also been shown to involve axonal and presynaptic components of neuronal transmission such as the positioning of the axon initial segment (AIS) and presynaptic transmitter release [10–12,41]. We now demonstrate that presynaptic mitochondrial occupancy may be another important factor during homeostatic rescaling.

Thus, we put forward a model whereby mitochondrial localisation at presynaptic terminals is tuned by neuronal activity and Miro1-mediated $Ca^{2+}$ sensing. This can increase mitochondrial occupancy when terminals are particularly active, thus enabling the mitochondria to provide energy and buffer $Ca^{2+}$ in those demanding conditions. Further, repositioning of mitochondria when network activity is altered on a longer timescale can contribute to the rescaling of presynaptic $Ca^{2+}$ signals during homeostatic plasticity.

## Materials and Methods

### Neuronal cultures and transfection

Primary hippocampal cultures were prepared as previously described from E16 mice [42], E18 rats or P0 rats [13,43]. Following 15 min (12 min for P0 rat) treatment with 0.25% trypsin and trituration, cells were plated on poly-L-lysine-coated, round, 12-mm coverslips for fixed experiments or 25-mm coverslips for live experiments at a density of 250,000 per 3-cm well. Neurons were transfected either by $Ca^{2+}$-phosphate precipitation or by lipofection with Lipofectamine 2000 at DIV 7 and then imaged at DIV 10–12.

### Antibodies, DNA constructs and reagents

MtDsRed, synaptophysin–GFP, myc-Miro1, myc-ΔEF-Miro1, VGlut1pHluorin and Mito-LSSmKate2 have been previously described [13,27,28]. The presynaptically targeted SyGCaMP5 was cloned using SyGCaMP2 (plasmid #26124 [17]) from Addgene as a target vector and inserting GCaMP5G from Addgene (plasmid #31788 [18]) via the restriction sites SalI and NotI [44]. ΔEF-Miro-IRES-MtDsRed was cloned with In-Fusion (Clontech) by inserting MtDsRed into pCAG-IRES-gfp using the BstXI and NotI sites. Then, myc-ΔEF-Miro1 was added using EcoRI and SacI sites. Picrotoxin (PTX) was purchased from Sigma-Aldrich and used at 100 μM, and tetrodotoxin (TTX) was purchased from Tocris Bioscience and used

at 1 μM. D(−)-2-Amino-5-phosphonopentanoic acid (APV) was purchased from Abcam and used at 100 μM. 1,2,3,4-Tetrahydro-6-nitro-2,3-dioxo-benzo[f]quinoxaline-7-sulphonamide (NBQX) disodium salt was purchased from Abcam and used at 10 μM. Ru360 was from Calbiochem and was used at 10 μM.

### Immunocytochemistry and fixed imaging

After fixation using 4% PFA for 5 min, cells were washed twice and blocked in PBS solution containing 10% horse serum, 0.5% BSA and 0.2% Triton. Cells were stained with primary antibody for 1 h: anti-myc antibody obtained from 9E10 hybridoma lines and used as supernatant at 1:100, anti-SV2 (Neuromab) used at 1:200, anti-Piccolo (Synaptic Systems) used at 1:500, anti-tau (Millipore) used at 1:1,000 and anti-MAP2 (Synaptic Systems) used at 1:1,000. Cells were washed and stained with secondary antibody for 1 h: anti-mouse 405 (Jackson Dylight) antibody was used at 1:500, anti-mouse Alexa 568 (Invitrogen) was used at 1:1,000, anti-rabbit Alexa 555 (Invitrogen) was used at 1:1,000 and anti-guinea pig Alexa 647 (Invitrogen) was used at 1:1,000. Images of fixed cultures were taken on a Zeiss LSM700 confocal using a 63× oil objective (NA 1.4) and a 20× water objective (NA 1.0).

### Live imaging

Imaging experiments were performed at 37°C while perfusing the coverslips in external solution containing 125 mM NaCl, 10 mM D-glucose, 10 mM HEPES, 5 mM KCl, 2 mM $CaCl_2$ and 1 mM $MgCl_2$, which was brought to a pH of 7.4 using NaOH. An inverted Zeiss Axiovert 200 microscope and a 63× oil objective (NA 1.4) coupled to a Photometrics Evolve camera were used to image frames with 30 ms exposure at either 1 frame per second or maximum frame rate (~18 frames per second) in the software Micro-Manager [45]. Using Chroma filters, coverslips were excited through a D470/40x filter and emission was split using an Opto-Split II (Cairn Research) [46] and a 565DCXR dichroic thereby collecting with HQ522/40M and HQ607/75M filters for SyGCaMP5 or VGlut1pHluorin and MtDsRed or Mito-LSSmKate2, respectively. Presynapses with and without mitochondria were discriminated on the basis of co-localisation of SyGCaMP5 or VGlut1pHluorin and MtDsRed or Mito-LSSmKate2, respectively: signal overlap for the duration of the imaging period was required for a synapse to be deemed occupied by a mitochondrion. Clear separation in signal was required to deem a synapse devoid of mitochondria. Field stimulation was achieved using a Grass S9 or S88 stimulator and a Warner Instruments stimulation bath. Individual stimulating pulses lasted for 1 ms and were set at 10 V as part of stimulation trains of variable frequencies (5–100 Hz) and durations (0.1–20 s). To reactivate neuronal firing, TTX-treated cultures were transferred to and washed in external solution for 20 min prior to imaging.

### Data analysis

Movies were aligned using the Cairn Image Splitter plugin in ImageJ. Graphs showing $\Delta F/F_0$ were plotted using *Mathematica* (Wolfram Research), and paired *t*-test was used to calculate statistical significance, whereby terminals with and without

mitochondria within the same axon were compared. Regions of interest were manually drawn, and after background subtraction, fluorescence was normalised to the first 10 frames. Mean stimulation fluorescence was calculated as an average across a plateau equating to stimulation duration. For co-localisation analysis, a region measuring $40 \times 40$ μm was chosen at least 300 μm from the soma. Co-localisation of SYN-GFP and MtDsRed was quantified as the fraction of SYN-GFP clusters which overlap with at least one MtDsRed-positive pixel. Images were thresholded in ImageJ and, using the Image Calculator tool, a third image was generated showing those pixels which were positive in both input channels. Using the Particle Analysis tool, the size and number of the thresholded clusters were analysed. Microsoft Excel was used to calculate the fraction of MtDsRed-positive SYN-GFP clusters. In order to quantify the density of mitochondria and SYN-GFP clusters within the axon, the whole axon was imaged using the 20× water-immersion objective. Images were stitched together in ImageJ, to reconstruct the whole axon, and the longest process was traced and straightened. The MtDsRed and SYN-GFP channels were thresholded and the number of thresholded clusters was analysed using the Particle Analysis tool within ImageJ. This was normalised to the straightened axon length. GraphPad Prism was used to perform ANOVAs and *t*-tests and to visualise bar charts. Error bars represent SEM.

Expanded View for this article is available online.

## Acknowledgements

We thank Julia Harris, Andrew MacAskill and David Attwell for reading earlier versions of this manuscript. V.V. was supported through the Wellcome Trust-funded 4-year UCL Neuroscience PhD programme. M.J.D. was supported by a Wellcome Trust Clinical Postdoctoral Fellowship (106713/Z/14/Z) and an Academy of Medical Sciences starter grant. N.H. was supported by an MRC studentship and Centenary Award. This work was further supported by a grant from the Wellcome Trust (093239/Z/10/Z), an ERC starting grant (Fuelling Synapses) and a research prize from the Lister Institute of Preventive Medicine to J.T.K.

## Author contributions

VV, MJD, NFH and JTK designed experiments; VV, MJD and NFH collected and analysed the data; NFH performed molecular biology experiments; and VV, MJD, NFH and JTK wrote the manuscript.

## Conflict of interest

The authors declare that they have no conflict of interest.

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
