## [Review Process File · EMBO Reports]

Manuscript EMBO-2016-42710

Miro1 dependent mitochondrial positioning drives the rescaling of presynaptic Ca²⁺ signals during homeostatic plasticity

Victoria Vaccaro, Michael J. Devine, Nathalie F. Higgs, and Josef T. Kittler

Corresponding author: Josef T. Kittler, University College London

Review timeline:

Submission date:	12 May 2016
Editorial decision:	14 June 2016
Revision received:	18 October 2016
Editorial decision:	07 November 2016
Revision received:	16 November 2016
Accepted:	28 November 2016

Transaction Report:

1st Editorial Decision

14 June 2016

Thank you very much for the submission of your research manuscript to our editorial office. We have now received the full set of reports from the referees that were asked to assess it.

While all reviewers agree in principle on the interesting nature of the observations reported here, they also point out a number of instances in which the data would need to be strengthened before publication. In particular, referee 3 feels that the data set is not yet sufficiently strong to support the idea that mitochondrial calcium buffering plays the proposed role in homeostatic synaptic plasticity. His/her main point, which is also brought up by referee 1, is that it should be tested at which stimulation threshold the mitochondria start to buffer calcium levels. S/he also feels that the localization of mitochondria in the presynaptic zone needs stronger experimental support and that controls need to be performed to analyze the effects of mitochondrial buffering on basal activities in the synapse. All reviewers also raise some other, rather minor issues, but I would like to refer you to their detailed reports for their specific requests.

Given the potential interest of your study and the constructive suggestions of the reviewers on how to improve it, I would like to give you the opportunity to revise your manuscript, with the understanding that the main concerns of the referees must be addressed and their suggestions taken on board. Acceptance of the manuscript will depend on a positive outcome of a second round of review and I should also remind you that it is EMBO reports policy to allow a single round of revision only and that, therefore, acceptance or rejection of the manuscript will depend on the completeness of your responses included in the next, final version of the manuscript.

Revised manuscripts should be submitted within three months of a request for revision; they will otherwise be treated as new submissions. If you feel that this period is insufficient for a successful submission of your revised manuscript I can potentially extend this period slightly.

REFEREE REPORTS

Referee #1:

The authors investigate here a very interesting issue, the level of calcium buffering undertaken by mitochondria in synaptic boutons. They analyze calcium levels, as well as synaptic vesicle release kinetics, in boutons with and without mitochondria, and come to the conclusion that mitochondria buffer some of the calcium entering the synapse during stimulation, and that this has functional consequences on vesicle release. Moreover, the authors show that the recruitment of mitochondria to synapses can be modulated by homeostatic changes in network activity, and that the recruitment requires a functional mitochondrial trafficking protein, Miro1, which is also a calcium sensor. While the suggestion that mitochondria calcium buffering has been introduced to the literature more than a decade ago, it has still remained largely uninvestigated. The study of the authors therefore is timely and interesting. I also find it convincing and well performed. One aspect, however, needs to be elaborated on more deeply: the level of calcium that is susceptible to mitochondria buffering. Mitochondria are typically not docked at synaptic active zones, and therefore synaptic release from docked and primed vesicles, which are found next to the calcium channels, is probably not affected by mitochondria buffering. I presume that mitochondria affect most the calcium concentrations away from the active zone, which may be involved more in recruitment of synaptic vesicles than in the process of exocytosis.

The authors may not be able to answer this type of speculation. Nevertheless, to guide future research on this subject, they should perform a more thorough analysis of the minimal stimuli that are not subjected to mitochondria buffering. The authors show in Figure 1 that responses to 1 AP stimulations are not affected. What about 2-10 APs? Are they affected? I would suspect that substantial differences would be observed between mitochondria-containing and mitochondria-free boutons only after the release of the entire primed pool of vesicles, which would happen somewhere between 10 and 40 APs (since ~40 APs are known to release the entire readily releasable pool, which contains the primed vesicles). A figure testing the SyGCaMP5 differences between mitochondria-containing and mitochondria-free boutons with different levels of stimulation (2 to 40 APs) would therefore be very helpful.

Minor points:

- 1) The cartoon in Figure 4G is not useful, and should therefore be removed.
- 2) The discussion could be condensed substantially, although this may be a matter of taste.

Referee #2:

The paper by Vaccaro et al. report the homeostatic induced change in mitochondrial occupation of presynaptic terminals. The authors used imaging methods to monitor individual synapses and their specific configuration in respect to the mitochondrial occupation and activity. the methods are adequate.

Major points:

The reported effects of induction of homeostatic plasticity by TTX treatment assume a major block of calcium influx through voltage gated calcium channels, since sodium channels are blocked. Another induction protocol for presynaptic homeostatic plasticity would be the use of postsynaptic block of AMPA and NMDAR. This would allow spontan action potentials to trigger voltage gated calcium channels. Would such stochastic action have the potential to trigger sufficient calcium influx to change the proposed trafficking of mitochondria via the MIRO1 action dependent on the EF-hand occupation by calcium?

The protocols to induce homeostatic plasticity are very standard, nevertheless testing mEPSC amplitude and frequency would still be an interesting piece of data, first as control for the induction protocols and second in respect to the shown alteration of mitochondrial occupation if the MIRO1 mutant. When MIRO1 Δ EF is coexpressed will this impact on the frequency of release? Such experiment could also support the argument whether the analysis of individual synapses will bring more mechanistic insight than global approaches, as mEPSC recordings are composed of many synapses.

Minor point: The age of used cultures should be announced

Referee #3:

In the present study, Vaccaro et al questioned whether neuronal activity regulates the distribution of mitochondria to active synapses in order to maintain proper Ca²⁺ homeostatic plasticity. Using live cell imaging of neuronal cultures transfected with SyGCaMP5 (a calcium indicator localized at presynaptic sites) and mitochondria markers, the authors show that mitochondria located at synapses buffer calcium and control neurotransmitter release. They further show that treatments with TTX (Na channel blocker) and picrotoxin (GABA_AR antagonist) affect the distribution of mitochondria at presynaptic sites, an effect that is mediated by the Ca²⁺-sensitive domain of Miro1. The overall idea that mitochondria are dynamically distributed to active synapses in order to control Ca²⁺ activity and neurotransmitter release is extremely elegant. However, most of the conclusions that are drawn from the current study are not supported by data and many experiments lack proper description and/or proper controls which hampers my original enthusiasm about this work. The following points should be addressed to support the conclusions and more controls are necessary to exclude any confounding factors.

Major:

1/ because the level of activity seems to be important to see the Ca²⁺ buffering effect by synaptic mitochondria (see Fig1B and D, 100AP vs 1AP), it would be interesting to know the threshold at which mitochondria start to play this role. Is this effect true for any kind of synaptic activity? Giving a range of activity (length of stimulation: 1 to 100 APs; or frequency of stimulation: 1 to 100Hz) would definitely strengthen the study and convince the reader that synaptic mitochondria play a role in homeostatic plasticity.

2/ In Fig4 the authors claim that mitochondria are dynamically distributed to active presynaptic sites using long term treatments of TTX and picrotoxin. I have many concerns about this set of experiments:

First the conclusions are not very convincing due to the low resolution of the confocal pictures. Based on the current staining it is very difficult to assess whether mitochondria localize at synapses or stay adjacent to these sites. New optical tools with better resolutions have provided many examples of misinterpretation of colocalization staining. Moreover the authors should describe how they discriminated mitochondria at a synapse versus mitochondria nearby synapses in the Methods section.

Second, the authors should provide live imaging showing the distribution of mitochondria and SYN-GFP before and after TTX and picrotoxin treatments to help better understand where these mitochondria come from. Is it a redistribution of a previously available and nearby pool? Also, is the overall number of axonal mitochondria affected in control vs TTX/PTX conditions? Is the number of presynaptic sites affected by long term treatments with TTX and PTX? In addition it would be interesting to know the dynamics of Ca²⁺-dependent mitochondria recruitment to active synapses in the context of homeostatic plasticity. Does this redistribution take minutes, hours or days?

Third, in the Miro-OE and Miro- Δ EF experiments, the authors do not provide any control experiments to show that overexpression of wild type Miro1 or Ca²⁺ insensitive Miro1 do not affect basal mitochondria trafficking, basal synaptophysin localization at synapses or normal formation of synapses. Also, how mitochondria are actively redistributed before and after treatments in these conditions is not tested.

Finally, Fig4G should be removed as it is not fully supported by data. Moreover the authors did not test whether the release of glutamate is higher at synapses devoid of mitochondria in TTX and PTX conditions.

3/ In Fig5, the authors treat hippocampal neurons with TTX for 48h and then apply a 10Hz field stimulation protocol to the culture which leads to increased SyGCamp5 signal at the synapse. This result is rather confusing given the fact that TTX inhibits the firing of action potentials in neurons by binding to voltage-gated sodium channels, in an irreversible manner. Therefore field stimulation of TTX-treated cultures should not lead to increased GCamp signal, unless this signal is unspecific to neuronal activity. It is nevertheless possible that field stimulation leads to Ca²⁺ influx directly into the synapse independent on neuronal activity. This point should be clarified

Minor:

- 1/ Fig EV1. Characterization of SyGCamp5 and localization at presynaptic terminals (SV2/SyGamp5 co-labeling). SV2 is mainly found on vesicles and is not specific for presynaptic terminals. The authors should perform new staining using other presynaptic markers such as SNAP25 or Bassoon to be convincing. The authors should also provide pictures of cell bodies and postsynaptic sites at dendrites to verify that the construct is specific for presynaptic sites. These controls are indispensable because SyGCamp5 is used as a central tool throughout the study.
- 2/ Neuronal cultures. Experiments were performed on E16 mouse neurons or P0 rat neurons without further indication of which cells were used in the figure legends. The authors should indicate the type of culture that was used in each experiment as it could affect the results. Also, the authors should indicate at what DIV the experiments were performed.
- 3/ The authors should provide a description for field stimulation protocols.
- 4/ In figures the n refers to the number of neurons that were analyzed. The authors could also provide the total number of terminals that were considered.

1st Revision - authors' response

18 October 2016

REFEREE COMMENTS

Referee #1:

The authors investigate here a very interesting issue, the level of calcium buffering undertaken by mitochondria in synaptic boutons. They analyze calcium levels, as well as synaptic vesicle release kinetics, in boutons with and without mitochondria, and come to the conclusion that mitochondria buffer some of the calcium entering the synapse during stimulation, and that this has functional consequences on vesicle release. Moreover, the authors show that the recruitment of mitochondria to synapses can be modulated by homeostatic changes in network activity, and that the recruitment requires a functional mitochondrial trafficking protein, Miro1, which is also a calcium sensor. While the suggestion that mitochondria calcium buffering has been introduced to the literature more than a decade ago, it has still remained largely uninvestigated. The study of the authors therefore is timely and interesting. I also find it convincing and well performed.

We would like to thank referee #1 for their supportive comments on our work.

One aspect, however, needs to be elaborated on more deeply: the level of calcium that is susceptible to mitochondria buffering. Mitochondria are typically not docked at synaptic active zones, and therefore synaptic release from docked and primed vesicles, which are found next to the calcium channels, is probably not affected by mitochondria buffering. I presume that mitochondria affect most the calcium concentrations away from the active zone, which may be involved more in recruitment of synaptic vesicles than in the process of exocytosis. The authors may not be able to answer this type of speculation. Nevertheless, to guide future research on this subject, they should perform a more thorough analysis of the minimal stimuli that are not subjected to mitochondria buffering. The authors show in Figure 1 that responses to 1 AP stimulations are not affected. What about 2-10 APs? Are they affected? I would suspect that substantial differences would be observed between mitochondria containing and mitochondria-free boutons only after the release of the entire

primed pool of vesicles, which would happen somewhere between 10 and 40 APs (since ~40 APs are known to release the entire readily releasable pool, which contains the primed vesicles). A figure testing the SyGCaMP5 differences between mitochondria-containing and mitochondria-free boutons with different levels of stimulation (2 to 40 APs) would therefore be very helpful.

We would like to thank referee #1 for their very insightful suggestion. We have performed an analysis of the number of stimuli that are subject to mitochondrial modulation, and find that 20 or more stimuli are required to see an effect (new Fig. 1F). This closely correlates with their prediction that the impact of mitochondria is expected to be seen only after the readily releasable pool is depleted.

Minor points:

1) The cartoon in Figure 4G is not useful, and should therefore be removed.

We have removed Fig 4G.

2) The discussion could be condensed substantially, although this may be a matter of taste.

In order to discuss our results in the context of previously published work, we have had to keep the discussion close to its original length.

Referee #2:

The paper by Vaccaro et al. reports the homeostatic induced change in mitochondrial occupation of presynaptic terminals. The authors used imaging methods to monitor individual synapses and their specific configuration in respect to the mitochondrial occupation and activity. The methods are adequate.

Major points:

The reported effects of induction of homeostatic plasticity by TTX treatment assume a major block of calcium influx through voltage gated calcium channels, since sodium channels are blocked. Another induction protocol for presynaptic homeostatic plasticity would be the use of postsynaptic block of AMPA and NMDAR. This would allow spontan action potentials to trigger voltage gated calcium channels. Would such stochastic action have the potential to trigger sufficient calcium influx to change the proposed trafficking of mitochondria via the MIRO1 action dependent on the EF-hand occupation by calcium?

We would like to thank referee #2 for their helpful comments and suggestions. As suggested we performed another presynaptic homeostatic plasticity protocol whereby glutamate receptors are blocked (APV: 100 μ M, NBQX 10 μ M, 24 hrs). This protocol also initiated the activity-dependent change in presynaptic mitochondrial occupancy (Fig EV4), an effect that was occluded following the overexpression of mutant Δ EF-Miro1 (Fig EV4).

The protocols to induce homeostatic plasticity are very standard, nevertheless testing mEPSC amplitude and frequency would still an interesting piece of data, first as control for the induction protocols and second in respect to the shown alteration of mitochondrial occupation if the MIRO1 mutant. When MIRO1 Δ EF is coexpressed will this impact on the frequency of release? Such experiment could also support the argument whether the analysis of individual synapses will bring more mechanistic insight than global approaches, as mEPSC recordings are composed of many synapses.

With reference to the suggestion to record mEPSCs from the cultured neurons in the presence of different Miro1 mutants, we feel that imaging has a major advantage over electrophysiology for two main reasons:

1) With imaging the behaviour of individual presynaptic terminals, and whether or not they are occupied by a mitochondrion, can be determined. In contrast, ensemble measurements via

electrophysiology would pool synapses both with and without mitochondria, potentially masking the contribution of mitochondria to presynaptic calcium signalling and vesicular release.

2) In addition, an electrophysiology experiment would be technically problematic to carry out. To determine the impact of a Miro mutant on presynaptic function electrophysiologically it would be necessary to record from an untransfected postsynaptic neuron receiving presynaptic inputs from a Miro mutant transfected neuron. Given we use sparse transfection it would be very difficult to identify such neurons. Moreover even if we could identify and record from a cell receiving inputs from a transfected cell, it would not be possible to guarantee that all the presynaptic inputs into the recorded cell would express the appropriate Miro1 construct (given that the recorded cell will also receive many inputs from non-transfected cells). Therefore the impact of manipulating Miro1 would be underestimated using this approach.

For these reasons we strongly believe that an imaging based method, allowing us to directly assess calcium signalling and transmitter release in neurons identified both for genetic manipulation and mitochondrial position, is the best approach.

Minor point: The age of used cultures should be announced.

We have updated the manuscript to include details of the age of cultures used in the experiments.

Referee #3:

In the present study, Vaccaro et al questioned whether neuronal activity regulates the distribution of mitochondria to active synapses in order to maintain proper Ca²⁺ homeostatic plasticity. Using live cell imaging of neuronal cultures transfected with SyGCaMP5 (a calcium indicator localized at presynaptic sites) and mitochondria markers, the authors show that mitochondria located at synapses buffer calcium and control neurotransmitter release. They further show that treatments with TTX (Na channel blocker) and picrotoxin (GABA_AR antagonist) affect the distribution of mitochondria at presynaptic sites, an effect that is mediated by the Ca²⁺-sensitive domain of Miro1. The overall idea that mitochondria are dynamically distributed to active synapses in order to control Ca²⁺ activity and neurotransmitter release is extremely elegant.

We would like to thank referee #3 for their complimentary remarks on our work.

However, most of the conclusions that are drawn from the current study are not supported by data and many experiments lack proper description and/or proper controls which hampers my original enthusiasm about this work. The following points should be addressed to support the conclusions and more controls are necessary to exclude any confounding factors.

We would like to thank referee #3 for their comprehensive and highly thoughtful series of comments and suggestions, which we address below.

Major:

1/ because the level of activity seems to be important to see the Ca²⁺ buffering effect by synaptic mitochondria (see Fig1B and D, 100AP vs 1AP), it would be interesting to know the threshold at which mitochondria start to play this role. Is this effect true for any kind of synaptic activity? Giving a range of activity (length of stimulation: 1 to 100 APs; or frequency of stimulation: 1 to 100Hz) would definitely strengthen the study and convince the reader that synaptic mitochondria play a role in homeostatic plasticity.

To address this point (which was also raised by referee #1), we have varied the number of stimuli to determine a threshold above which mitochondria are able to modulate presynaptic calcium responses and thus vesicular release. We show that 20 or more stimuli are required to see an effect, which likely correlates with near or complete depletion of the readily releasable pool. Interestingly, varying AP frequency appears not to have an additional effect. For example, increasing frequency of delivery of 10 stimuli from 10 to 100 Hz does not trigger a mitochondrial effect (which we hypothesised might be the case, if rate of calcium rise at the presynapse is important). In contrast, a

mitochondrial effect is still seen when 20 stimuli are delivered at a range of frequencies, from 5 Hz to 100 Hz.

2/ In Fig4 the authors claim that mitochondria are dynamically distributed to active presynaptic sites using long-term treatments of TTX and picrotoxin. I have many concerns about this set of experiments:

First the conclusions are not very convincing due to the low resolution of the confocal pictures. Based on the current staining it is very difficult to assess whether mitochondria localize at synapses or stay adjacent to these sites. New optical tools with better resolutions have provided many examples of misinterpretation of colocalization staining. Moreover the authors should describe how they discriminated mitochondria at a synapse versus mitochondria nearby synapses in the Methods section.

The aim of our study was to determine the functional impact of mitochondria on the presynapse by analysing the difference between boutons with and without mitochondria. For this purpose our imaging approach is appropriate, because we can readily detect terminals that contain or lack a mitochondrion and we do indeed detect differences in calcium signals and neurotransmitter release between them. Although interesting questions, our aim was not to define the precise distance (i.e. with nanometer resolution) between mitochondria and the active zone of presynapses, nor to determine the distance over which mitochondria can influence presynaptic function. Therefore our opinion is that, while additional imaging techniques may be informative in future studies, they are not critical to support the main conclusion of our study which is that mitochondrial occupancy of a presynaptic terminal can impact on presynaptic function.

We have updated the manuscript to include a description of how synapses with and without mitochondria are discriminated, in Materials and Methods. Briefly, overlap in fluorescent signal from SyGCaMP5 and MtDsRed, or VGlut1pHluorin and LSS-mKate, for the duration of the imaging period is required to deem a presynaptic bouton to contain a mitochondrion, whilst clear separation in signal is required to deem it to be without a mitochondrion.

Second, the authors should provide live imaging showing the distribution of mitochondria and SYN-GFP before and after TTX and picrotoxin treatments to help better understand where these mitochondria come from. Is it a redistribution of a previously available and nearby pool? Also, is the overall number of axonal mitochondria affected in control vs TTX/PTX conditions? Is the number of presynaptic sites affected by long-term treatments with TTX and PTX? In addition it would be interesting to know the dynamics of Ca²⁺-dependent mitochondria recruitment to active synapses in the context of homeostatic plasticity. Does this redistribution take minutes, hours or days?

In order to assess whether the number of axonal mitochondria or presynaptic sites was affected by the long-term treatment of TTX or PTX, we calculated the number of mitochondria or SYN-GFP clusters within the axon, for each condition. As shown in Figure EV3 A & B there is no significant difference in the density of mitochondria or SYN-GFP clusters within the axon following treatment. This agrees with previously published data from [1], which is in keeping with a redistribution of a previously available nearby pool of mitochondria within the axon.

The previous report by [1] also measured the transition of mitochondria from the mobile to the stationary state and also the stationary to the mobile state following TTX treatment. When they imaged over 3 hours, the majority of axonal mitochondria imaged at the initial time point remained stationary throughout the experiment. It was only following imaging at 1 day intervals over a 4 day period did they observe movement of the majority of mitochondria, following TTX treatment. This implies that the redistribution of mitochondria is within the region of many hours to days.

Third, in the Miro-OE and Miro-ΔEF experiments, the authors do not provide any control experiments to show that overexpression of wild type Miro1 or Ca²⁺ insensitive Miro1 do not affect basal mitochondria trafficking, basal synaptophysin localization at synapses or normal formation of synapses. Also, how mitochondria are actively redistributed before and after treatments in these conditions is not tested.

Overexpression of both wild type *Miro1* and Ca^{2+} -insensitive *Miro1* increase basal mitochondrial trafficking in a similar manner and has been reported previously by [2]. To investigate whether this overexpression affects basal synaptophysin localisation and synapse formation, we have quantified the density of SYN-GFP clusters along the axon and find that there is no difference in the density of SYN-GFP clusters per μm in control versus *Miro1* OE and *Miro1* ΔEF cells. We also show that the SYN-GFP clusters still colocalise with the presynaptic marker Piccolo (see Figure EV4D & E).

Finally, Fig4G should be removed, as it is not fully supported by data.

We have removed this figure panel from the manuscript.

Moreover the authors did not test whether the release of glutamate is higher at synapses devoid of mitochondria in TTX and PTX conditions.

We have analysed vesicular release at presynapses with and without mitochondria (via *VGlut1pHluorin* and *Mito-LSSmKate2* imaging) following 48 hours treatment with PTX and we have included this data as additional panels in Fig EV3 (C&D). This analysis confirms that vesicular release is indeed higher in synapses devoid of mitochondria following PTX treatment, in agreement with data shown in Fig 3B. This supports one of the main conclusions of the manuscript, that mitochondrial relocation can be a major component of synaptic rescaling. For completion, we also carried out the same analysis in neurons following 48 hours treatment with TTX, shown here:

In the left panel, average traces are shown from presynaptic terminals of hippocampal neurons transfected with *VGlut1pHluorin* and *Mito-mKate2* and stimulated using field stimulation at 10 Hz for 20 s (40-60s). Terminals with a mitochondrion are represented by the red trace and terminals without are represented by the black trace. In the right panel, a summary graph corresponds to the traces shown on the left. Mean stimulation fluorescence (time points 60-70s) is compared in terminals of the same neuron which contain mitochondria (right) compared to terminals without mitochondria (left) ($n=13$ neurons, 41 terminals, paired t -test, $*p<0.05$). Again, presynapses devoid of mitochondria exhibit higher vesicular release under the same stimulation conditions following 48h treatment with TTX.

3/ In Fig5, the authors treat hippocampal neurons with TTX for 48h and then apply a 10Hz field stimulation protocol to the culture which leads to increased SyGCamp5 signal at the synapse. This result is rather confusing given the fact that TTX inhibits the firing of action potentials in neurons by binding to voltage-gated sodium channels, in an irreversible manner. Therefore field stimulation of TTX-treated cultures should not lead to increased GCamp signal, unless this signal is unspecific to neuronal activity. It is nevertheless possible that field stimulation leads to Ca^{2+} influx directly into the synapse independent on neuronal activity. This point should be clarified.

Tetrodotoxin is a reversible inhibitor of voltage-gated sodium channels [3]. Importantly, prior to imaging neuronal response to field stimulation, the cultures must be perfused with external solution for a sufficient period of time to wash off residual TTX and to allow the cultures to recover and to be

able to fire action potentials. This takes approximately 10-20 mins [4,5]. We have clarified this point in Materials and Methods.

Minor:

1/ Fig EV1. Characterization of SyGCamp5 and localization at presynaptic terminals (SV2/SyGamp5 co-labeling). SV2 is mainly found on vesicles and is not specific for presynaptic terminals. The authors should perform new staining using other presynaptic markers such as SNAP25 or Bassoon to be convincing. The authors should also provide pictures of cell bodies and postsynaptic sites at dendrites to verify that the construct is specific for presynaptic sites. These controls are indispensable because SyGCamp5 is used as a central tool throughout the study.

In order to confirm the presynaptic localization of SyGCAMP5 we have performed new staining of SyGCAMP5 showing its colocalisation with the presynaptic marker Piccolo (see Figure EV1C). We have also included pictures of SyGCAMP5, co-stained with Tau (axonal marker) and MAP2 (dendritic marker) to confirm that SyGCAMP5 is found within the axon specifically (see Figure EV1D).

2/ Neuronal cultures. Experiments were performed on E16 mouse neurons or P0 rat neurons without further indication of which cells were used in the figure legends. The authors should indicate the type of culture that was used in each experiment as it could affect the results. Also, the authors should indicate at what DIV the experiments were performed.

We have added this information to the figure legends, and have also clarified the Materials and Methods accordingly.

3/ The authors should provide a description for field stimulation protocols.

We have added this information to the manuscript.

4/ In figures the n refers to the number of neurons that were analyzed. The authors could also provide the total number of terminals that were considered.

We have added these numbers to the manuscript, for completion.

References:

1. Obashi K, Okabe S (2013) Regulation of mitochondrial dynamics and distribution by synapse position and neuronal activity in the axon. *Eur J Neurosci* **38**: 2350–2363.
2. MacAskill AF, Rinholm JE, Twelvetrees AE, Arancibia-Carcamo IL, Muir J, Fransson A, Aspenstrom P, Attwell D, Kittler JT (2009) Miro1 Is a Calcium Sensor for Glutamate Receptor-Dependent Localization of Mitochondria at Synapses. *Neuron* **61**: 541–555.
3. Narahashi T, Moore JW, Scott WR (1964) Tetrodotoxin Blockage of Sodium Conductance Increase in Lobster Giant Axons. *J Gen Physiol* **47**: 965–974.
4. Zhao C, Dreosti E, Lagnado L (2011) Homeostatic synaptic plasticity through changes in presynaptic calcium influx. *Journal of Neuroscience* **31**: 7492–7496.
5. Nakayama K, Kiyosue K, Taguchi T (2005) Diminished neuronal activity increases neuron-neuron connectivity underlying silent synapse formation and the rapid conversion of silent to functional synapses. *Journal of Neuroscience* **25**: 4040–4051.

Thank you for the submission of your revised manuscript to EMBO reports. We have now received the full set of referee reports that is copied below. As Barbara Pauly has left EMBO reports in July, I am the new handling editor of your manuscript. As you will see all three referees are very positive about the study and support its publication in EMBO reports. There are now only a few things that we need from the editorial side before we can proceed with the official acceptance of your study.

- You have submitted your manuscript as Scientific Report. In this case the Results and Discussion sections have to be combined. This will help to shorten the manuscript text by eliminating some redundancy that is inevitable when discussing the same experiments twice. The main text may have 25,000 (plus minus 2,000) characters - references and materials & methods are excluded from this limit.
- Regarding data quantification, can you please specify the error bars (SEM) also in the figure legends?
- Please ensure that the resolution of all figures files is at least 300 ppi at final print size.

REFEREE REPORTS

Referee #1: The authors have addressed my comments, and I am now happy to suggest that the manuscript be published.

Referee #2: The authors responded to all of my comments adequately. The proposed functional aspects on homeostatic plasticity are sound and well explored. I have no further comments.

Referee #3: Authors have answered most of my comments, the paper can now be considered suitable for publication.

2nd Revision - authors' response

16 November 2016

The authors made the requested changes and submitted the final version of the manuscript.

3rd Editorial Decision

28 November 2016

I am very pleased to accept your manuscript for publication in the next available issue of EMBO reports. Thank you for your contribution to our journal.

YOU MUST COMPLETE ALL CELLS WITH A PINK BACKGROUND

Corresponding Author Name: Josef Kittler
 Journal Submitted to: EMBO Reports
 Manuscript Number: EMBO-2016-42710V2